# Influence of Modification of the Plasticizing System on the Extrusion-Cooking Process and Selected Physicochemical Properties of Rapeseed and Buckwheat Straws

**DOI:** 10.3390/ma15145039

**Published:** 2022-07-20

**Authors:** Karol Kupryaniuk, Tomasz Oniszczuk, Maciej Combrzyński, Katarzyna Lisiecka, Damian Janczak

**Affiliations:** 1Department of Thermal Technology and Food Process Engineering, University of Life Sciences in Lublin, Głęboka 31, 20-612 Lublin, Poland; karol.kupryaniuk@o2.pl; 2Department of Biochemistry and Food Chemistry, University of Life Sciences in Lublin, Skromna 8, 20-704 Lublin, Poland; katarzyna.lisiecka@up.lublin.pl; 3Institute of Biosystems Engineering, Poznań University of Life Sciences, Wojska Polskiego 50, 60-627 Poznań, Poland; janczak.dam@gmail.com

**Keywords:** thermomechanical pretreatment, biogas, methane, biomass, lignocellulose materials

## Abstract

The article discusses the effect of modification of the plasticizing system of a single-screw extruder on selected physicochemical properties of rapeseed straw and buckwheat straw. A TS-45 single-screw extruder (ZMCh Metalchem, Gliwice, Poland) with an L/D = 12 plasticizing system was used for the process. The shredded straws were moistened to four moisture levels: 20, 25, 30 and 35% dry matter. Three different rotational speeds of the extruder screw were applied for the test cycle: 70, 90 and 110 rpm. The following characteristics were determined for the extrusion-cooking process: efficiency and specific mechanical energy. Selected physical properties were determined for the extrudates obtained in the process: water absorption index (WAI), water solubility index (WSI), bulk density, and the efficiency of cumulative biogas and cumulative methane production expressed on dry mass, fresh mass, and fresh organic mass basis. It has been proved that the modification of the plasticizing system had a significant impact on the course of the process and the tested physicochemical properties. An important factor confirming the correctness of the modification is the increase in biogas efficiency. After modification, the highest yield of cumulative biogas from the fresh mass was 12.94% higher than in the sample processed before modification.

## 1. Introduction

A developing society needs energy to sustain its growth. Growing consumerism, technological progress, volatile geopolitical situations, challenges posed by the supply of fossil fuels, continuous population growth, production of foodstuffs and other items of everyday life—all these drive up the global energy demand [1]. Environmental protection has become the top priority, and sustainable waste management at various levels allows the reduction in biosphere pollution. Agricultural and forest waste, as well as waste from the agri-food industry, are becoming a more and more burning issue. Ligno-cellulosic materials, such as hay or straw, are not only intended for animal nutrition but are often considered farm waste [2]. Energy transition and problems with the supply of fossil raw materials make their use for power generation purposes more than justified. Renewable energy has become the core element of sustainable development nowadays. The ecological benefits and the prevention of global warming are important, yet not the only advantages of RES (Renewable Energy Sources). Some others include access to nearby energy resources, new jobs, the country’s proper energy balance, or becoming independent of energy imports [3]. By laying down new requirements limiting the emission of harmful gases from the burning of fossil fuels, the European Union has obliged the member states to introduce new, innovative technologies to increase the use of RES and control the consumption of cleaner energy in Europe. The laws made by the Polish legislator, in particular on extra financing options, have raised the share of RES in the national energy balance [4]. However, the mere interest in RES does not suffice; therefore, the various types of available funding are aimed at transforming the curiosity and growing awareness of the general public into the practical use of renewable energy. The beneficiaries of green electricity programs can be companies, private individuals, and also farmers. The funding is sourced from, for example, the EU, domestic budgets or the Green Investment Scheme (GIS). The money supports many programs with specific beneficiaries, the nature of funding, or the conditions and indicators that must be met (e.g., to generate a certain amount of electrical energy per year). Some projects that have gained popularity in Poland and have created public interest are, for example, PROSUMENT, BOCIAN, ROP (Regional Operational Programs), Energia Plus [5]. One of the ways of managing green energy is its use in agricultural biogas plants. Waste plant raw materials can be successfully used as a source of energy.

Biogas is a product of anaerobic digestion. Adequate hydration and temperature cause the development of bacteria that feed upon organic matter. The product of this process is combustible biogas, also known as mud/marsh gas [6]. Agricultural biogas is produced from raw materials of animal and plant origin. Animal raw materials include manure and slaughter waste (liquid manure, slurry, and stable manure). Plant substrates are energy crops, food and garden waste, and grass clippings. The main substrate used in biogas plants is maize silage. It is diluted in slurry, which stabilizes the process of methane fermentation [7,8]. Energy crops used as silage, whole plants, tubers or seeds represent a wide group covering broad bean, grass, rapeseed, buckwheat, oats, mustard, beetroot, etc. [9]. The most commonly used raw material in the production of agricultural biogas in Poland is maize silage and slurry. Comparing the number of agricultural biogas plants operating in Poland (117) and in Germany (about 10,000), a threat to the food industry is far from real; still, the number of such installations is growing, and the range of species providing an input for biomass is broad, similarly to the volume of waste generated by the agri-food industry [10]. Post-harvest waste of rapeseed includes roots, siliques, dried leaves and stems, which, when left in the field, turn into natural fertilizer. Straw improves the soil structure by supplying micro- and macronutrients [11]. Rapeseed straw consists of a hard, hemicellulose- and lignin-rich epidermis and a spongy interior, largely of cellulose [12]. Buckwheat straw has a high content of volatile components and a large amount of oxygen. As a result, it has a very high concentration of released heat energy in the initial stage of the incineration process. For energy-generation purposes, grey straw is most often used, i.e., one left after harvest in the field for several days to enable rainwater to leach chlorine and potassium compounds from it. Thanks to this simple procedure, the corrosive effect of exhaust gases on the boiler surfaces is limited [13]. As reported [14], the most fertile soils in Poland (chernozem) account for less than 0.75% of arable land; therefore, in order to avoid fertilization of poorer-quality lands, pressure is put on energy crops that do not require high-quality soil.

According to the available literature [15,16], there are several methods of pretreatment of lignocellulose raw materials used to produce energy. However, it has not been determined yet which of them is the most efficient. Selection of the best available technology depends on various factors such as the type and size of biomass or the target product to be obtained as a result of processing [17]. There are several basic methods of pretreatment, among them microbiological processing, which uses microorganisms occurring in natural conditions. Another method is chemical pretreatment, in which the action of chemicals results in preliminary hydrolysis. However, the latter requires special (non-metallic) tanks and that acid be recovered after the process has been completed. Among the mechanical treatment methods, there is micronization. It is commonly employed in the production of animal feed or biomass combustion. This method mainly proves effective with raw materials for the production of silage. One of the methods of pressure-thermal treatment is extrusion-cooking. Due to the extensive process configuration options, it can be adapted to a wide range of biomass inputs [18]. The application of single- or twin-screw extruders may have an impact on obtained results. The influence of high pressure, temperature and the rotation speed of the extruder screws causes lignin bonds to break and releases cellulose and hemicelluloses, which increases the surface of active biomass and initiates hydrolysis. The extrusion-cooking process can be combined with other pretreatment methods, for example, the above-mentioned micronization. There are a number of factors that determine the ultimate effect of extrusion-cooking. These are, among others, the degree of raw material fragmentation and its moistening, the appropriate process temperature, and the choice of the proper extruder plasticizing system [19]. The extrusion-cooking process is energy intensive, but under certain conditions, it can be very productive and cost-effective. The extruder cylinder can be heated using the waste heat from a biogas plant or excess generated electricity, which significantly reduces the costs of the entire process. The lignocellulosic raw material after extrusion can be stored and used at any time during the production of biogas. Owing to extrusion-cooking, the input material for a biogas plant can be supplied on a continuous basis, and the temperature ranging from 130 to 180 °C sufficiently protects the product against the development of microorganisms [20].

The aim of the research was to determine the influence of modification of the plasticizing system on the course of the extrusion-cooking process and on the physicochemical properties of selected pretreated straws as well as the effect of pretreatment on biogas efficiency during methane fermentation.

## 2. Materials and Methods

As the main raw materials, rapeseed straw and buckwheat straw were used. The rape straw used in the tests was qualitatively assessed by 4.76% ash content, 19.41% lignin content, 41.33% cellulose content, 30.57% hemicellulose content, 42.47% carbon content and 0.51% nitrogen content. In the case of buckwheat straw, there was 5.73% ash content, 19.94% carbon content, 36.54% cellulose content, 28.26% hemicellulose content, 37.28% carbon content and 1.17% nitrogen content [21]. The studies were carried out on extrudates made of rapeseed and buckwheat straws at three rotational speeds of the extruder screw (70, 90, and 110 rpm). The raw materials used were shredded to a size of 8–10 mm and moistened with tap water to the initial moisture content of 20, 25, 30, and 35% dry mass. A moisture analyzer (Radwag, MA 50.R.WH, Radom, Poland) was used to determine the initial moisture content of the straws.

A TS-45 single-screw extruder (Metalchem, Gliwice, Poland) with an L/D = 12 plasticizing system was used for the process. A standard and a prototype screw were used. It was equipped with an additional section of beaters to ensure better compaction of the processed raw material before its molding phase at the extruder head (Figure 1).

During the treatment of straw, the processing efficiency and specific mechanical energy consumption of the extrusion-cooking process were determined, and the influence of process variables on selected physical and chemical properties of lignocellulosic raw materials was tested with the application of two different plasticizing systems (before and after modification). The followed tests also determined the effect of extruder screw modification (making additional cuts in the mixing zone) and process variables on the biogas efficiency of tested pretreated straws (Figure 2).

### 2.1. Extrusion-Cooking Efficiency

The testing of the efficiency of the extrusion-cooking process focused on determining mass obtained over a specific time for all the raw materials used and at pre-set process parameters. The measurements were performed three times for each test series. The result was the mean value of all the measurements. The processing efficiency was determined with the formula:(1)$$Q=\;\frac{m}{t}\;(\mathrm{kg}\;h^{-1})$$
where:Q—process efficiency (kg h^−1^),m—mass obtained during measurement (kg),t—measurement time (h).

### 2.2. Specific Mechanical Energy Consumption of Extrusion-Cooking

Energy consumption of the extrusion-cooking process was determined based on the specific mechanical energy (SME) calculated according to the following formula [22]:(2)$$\mathrm{SME}=\;\frac{n}{N}\frac{L}{100}\frac{P}{Q}\;(\mathrm{kWh}\;{\mathrm{kg}}^{-1})$$
where:SME—specific mechanical energy (kWh kg^−1^),n—extruder screw speed (rpm),N—maximum extruder screw speed (rpm),L—motor load (%),P—power (kW),Q—process efficiency (kg h^−1^).

### 2.3. Measuring WAI (Water Absorption Index)

The processed straw samples obtained through extrusion were ground with a laboratory grinder (ELDOM MK100S, Katowice, Poland) into particles with a diameter not greater than 0.8 mm. A suspension was prepared from a sample of 0.7 g and 7 mL of distilled water by continuous mixing for 20 min. The suspension was centrifuged at 15,000 rpm for 10 min in a laboratory centrifuge (Digicen 21, Madrit, Spain). Filtrate was collected from the obtained gel, and next, the gel was weighed [23]. The water absorption index was calculated using the formula:(3)$$\mathrm{WAI}=\frac{m_{z}}{m_{s}}\;(g\;g^{-1})$$
where:WAI—water absorption index (g g^−1^),m_s_—dry sample mass (g),m_z_—gel mass (g).

### 2.4. Measuring WSI (Water Solubility Index)

The filtrate obtained during the measurement of the WAI was dried at 130 °C until total water evaporation [24]. The water solubility index was calculated according to the formula:(4)$$\mathrm{WSI}=\frac{m_{s}-m_{\mathrm{ps}}}{m_{\mathrm{pp}}}\;100\;(\%)$$
where:WSI—water solubility index (%),m_s_—vessel mass after drying (g),m_ps_—vessel mass before drying (g),m_pp_—sample mass (g).

### 2.5. Measuring Bulk Density of Processed Straws

Density was measured by pouring the tested material through a funnel into a 500 cm^3^ vessel (any excess material was swept in as well) and then by weighing the vessel with the content. Based on the obtained mass, the bulk density was calculated as follows [25]:(5)$$\rho_{u}=\frac{m_{u}}{v}\;\left({\mathrm{kg}\;m}^{-3}\right)$$
where:$\rho_{u}$—bulk density (${\mathrm{kg}\;m}^{-3}$),m_u_—sample mass (kg),v—vessel volume (m^3^).

### 2.6. Methane Fermentation of Processed Raw Materials

The study was conducted at the Laboratory of Ecotechnologies of the Poznan University of Life Sciences (PULS). Biogas efficiency was investigated in the methane fermentation process under mesophilic conditions (the most popular technology in Europe) in three replications with the use of proprietary biofermentors [26] (Figure 3).

The pretreated with the extrusion-cooking samples were analyzed for biogas efficiency in accordance with the generally recognized standards, namely DIN 38414/S8 and VDI 4630 [27]. The tests were carried out in mesophilic conditions in sets of 3-vessel biofermentors [28]. Fermentation reactors with a capacity of 2 dm^3^ were filled with inoculum (with a dose of microorganisms from an operating biogas plant) and with rapeseed and buckwheat straws processed under various conditions of the extrusion-cooking process. The content of organic dry matter in the inoculum ranged from 1.5 to 2%. Dry mass and organic dry mass were measured prior to testing. After that, the substrates were placed in an airtight reactor for fermentation. The vessels with samples were immersed in water at a controlled temperature (about 39 °C), which simulated the actual operating conditions of biogas installations. The volume and qualitative composition of generated gases were measured every 24 h. The fermentation process was stopped when the daily biogas production lowered below 1% of the total amount of biogas produced. The samples were tested in three replications. The biogas yield was measured (m^3^ Mg^−1^) relative to fresh mass (rapeseed and buckwheat straws), dry mass, and dry organic mass, as described by Dach et al. [29].

### 2.7. Analysis of Results

The data obtained during the tests were archived and analyzed statistically using Microsoft Office Professional Plus 2019 (Microsoft, Redmond, WA, USA, Excel) and Statistica 13.3 (TIBCO Software Inc., Palo Alto, CA, USA). The RSM method with surface approximation was used to investigate the dependence of specific results on variable process parameters.

The similarity of the screw systems (before and after modification) was analyzed using the principal component analysis (PCA) with Statistica 13.3. The PCA data matrix for the statistical analyses of both types of extrudates results had 11 columns and 24 rows. The input matrix was scaled automatically. The correct number of main components obtained in the analysis was determined based on Cattell’s criterion.

## 3. Results and Discussion

The results shown below demonstrate the impact of modification of the plasticizing system of a single-screw extruder and of process variables on the extrusion-cooking and selected physicochemical properties of rapeseed straw and buckwheat straw. The results are presented both in tables and figures.

### 3.1. Extrusion-Cooking Efficiency

The efficiency of the extrusion-cooking process depends on many factors: the rotational speed of the extruder screw, the type and moisture content of the raw material mixture used in the process, or the configuration of the extruder screws [19]. The chapter below discusses the test results of the efficiency of extrusion-cooking of lignocellulosic material, i.e., buckwheat and rapeseed straws.

Figure 4 shows the results of measurements of the efficiency of the extrusion-cooking of rapeseed and buckwheat straws when a standard extruder screw was used. For rapeseed straw (Figure 4A), higher process efficiency was observed to be correlated with the higher rotational speed of the extruder screw. The highest efficiency (33.12 kg h^−1^) was reported for straw extruded at 25% initial moisture and the screw rotational speed of 110 rpm. The lowest efficiency of rapeseed straw (5.38 kg h^−1^) was reported for one extruded at the screw speed of 70 rpm and at a moisture content of 35%. A considerably higher efficiency was measured in extrusion trials at 25 and 30% moisture levels than for 20 and 35%. The initial increase in the moisture content from 20 to 25% resulted in a significant surge in efficiency. However, when the moisture level kept increasing to 30%, the efficiency began to drop. Similar extrusion tests were carried out with maize straw [30]. The moisture content of 50% and the screw speed of 90 rpm delivered a process efficiency of 18.24 kg h^−1^, which is close to the effect achieved when using a rapeseed straw at a 30% level of moisture and the screw speed of 90 rpm (19.44 kg h^−1^).

In the case of buckwheat straw, the highest efficiency (11.04 kg h^−1^) was reported at 20% moisture and at the screw speed of 90 rpm during processing. The lowest efficiency for this input material (Figure 4B) (4 kg h^−1^) was observed at 25% moisture and the screw speed of 110 rpm. It has been observed that as the moisture level increases, the efficiency of the extrusion-cooking process of the buckwheat straw decreases. Depending on the level of moisture, the highest efficiency was obtained at different rotational speeds of the extruder screw. The process with the raw material moistened up to 25% showed the highest efficiency at the screw speed of 70 rpm (8.08 kg h^−1^), while at 30% (8.24 kg h^−1^) and 35% (4.96 kg h^−1^) moisture, the maximum efficiency was recorded at 110 rpm.

Figure 5 shows the results of the measurement of the efficiency of the extrusion-cooking process of rapeseed and buckwheat straws after the modification of the plasticizing system. As regards buckwheat straw (Figure 5B), the lowest levels of the measured quality were found in samples processed at the lowest screw speeds. The lowest efficiency (8.64 kg h^−1^) was recorded for samples processed at the screw speed of 70 rpm and at 20% initial moisture. The higher efficiency (apart from at the moisture content of 20%) was seen in samples processed at the highest rotational speed. The maximum result (17.68 kg h^−1^) was recorded when applying a 30% moisture level. Compared with the standard (non-modified) screw, the modification of the plasticizing system increased the extrusion-cooking efficiency across the entire range of applied process variables.

In the case of rapeseed straw extrusion-cooking (Figure 5A), much more promising measurements were recorded for 35% moisture across the entire range of the screw speed. The lowest result obtained (5.88 kg h^−1^) was recorded at the moisture level of 25% and the screw speed of 90 rpm. The highest efficiency (20.16 kg h^−1^) was measured during processing a sample with 35% initial moisture and at the screw speed of 110 rpm. Much higher efficiency results were recorded in samples processed at the moisture content of 30 and 35%; when the standard screw was used, the maximum efficiency values were recorded for samples processed with 25 and 30% moisture.

Table 1 contains equations showing the functions of matching the response surface model for the examined feature based on the variables used.

### 3.2. Energy Consumption during Processing

The extrusion-cooking process is energy intensive. The appropriate selection of the parameters of the extrusion-cooking process, in particular the type of and the moisture content in extruded raw material mixtures, significantly affects the volume of energy consumed by the process. Modifying the moisture content in the mixture by even 1–2% may increase the energy intensity of the process by up to several dozen percent [31]. The economic success of this technique depends on the choice of optimal parameters of extrusion-cooking, in particular the moisture of raw material mixtures, and adequate design modifications of the extruder plasticizing system.

The energy consumption of the extrusion-cooking of rapeseed and buckwheat straws reveals the relationship between the moisture content and the rotational speed of the extruder screw (Figure 6). For rapeseed straw processed with the standard plasticizing system (Figure 6A), at the screw speeds of 70 and 90 rpm, the energy intensity of the process gradually decreased along with the increasing moisture content from 20 to 25%; upon a further increase to 30% moisture, the energy consumption began to rise again. In the case of the screw speed of 110 rpm, the rising moisture content from 20 to 25 and 30% caused a reduction in energy intensity; a continued increase in the moisture content (up to 35%) resulted in a greater energy consumption again. The lowest SME in the rapeseed straw extrusion-cooking process (0.165 kWh kg^−1^) was calculated for samples pretreated at 25% moisture and the screw speed of 90 rpm. The highest SME value (0.686 kWh kg^−1^) was obtained in samples extruded at 35% moisture and the screw speed of 90 rpm.

In the case of extrusion-cooking of buckwheat straw (Figure 6B), at the screw speed of 70 rpm, the energy consumption of the pretreatment process increased along the entire growing range of the moisture content. At the speeds of 90 and 110 rpm, along with the rise of the moisture level from 20 to 25%, there was a gradual increase in energy consumption, followed by a drop upon the moisture level reaching 30%. A further growth in the moisture content to 35% again put up the energy consumption value. Regardless of the moisture level, each increase in the rotational speed of the extruder screw led to an increase in SME. The lowest energy was required during buckwheat straw extrusion-cooking processing (0.273 kWh kg^−1^) for samples processed at 20% moisture and the screw speed of 70 rpm. The highest SME value (1.432 kWh kg^−1^) was calculated in samples processed at 25% moisture and the screw speed of 110 rpm. Roye [32] conducted research on the energy intensity of wheat bran extrusion-cooking. In this case, energy consumption ranged from 0.026 to 0.046 kWh kg^−1^. Moreover, in their work, Pardhi et al. [33] shared an analysis of the energy consumption of brown rice extrusion-cooking. For this raw material, the energy demand ranged from 0.101 to 0.138 kWh kg^−1^. In both cases, the energy consumption measurements presented by the mentioned authors showed lower values than obtained during the pretreatment of the tested straws.

Figure 7 shows the results of SME for the extrusion-cooking of rapeseed and buckwheat straws and reveals the relationship between the moisture content and the rotational speed of the prototype extruder screw. Lower process energy requirements were calculated during the extrusion of buckwheat straw (Figure 7B). The lowest value (0.181 kWh kg^−1^) was obtained in the case of a sample extruded at 25% moisture and the screw speed of 70 rpm. The highest SME was reported in samples processed at the screw speed of 110 rpm across the entire range of the moisture content, and the top energy consumption value (0.474 kWh kg^−1^) was calculated for a sample processed at a 20% level of moisture.

In the case of rapeseed straw (Figure 7A), pretreated at the screw speeds of 70 and 90 rpm, it was observed that the initial increase in the moisture content from 20 to 25% raised the energy intensity of the process, but at further process stages, the SME began to fall. At the screw speed of 110 rpm, a higher moisture content (up to 25%) caused a minor decrease in energy intensity; however, upon the subsequent increase in the level of moisture (up to 30%), energy intensity began to subside. Finally, raising the moisture level to 35% decreased the SME value. The highest value (0.962 kWh kg^−1^) was calculated for a sample processed at 25% moisture and at the screw speed of 90 rpm. The lowest result (0.223 kWh kg^−1^) was reported at the moisture level of 35% and the screw speed of 70 rpm.

When using the standard screw, the energy requirements of the rapeseed straw extrusion-cooking process were higher for the samples processed at the top level of moisture (35%); in other cases (20–30%), higher values were reported for the samples processed with the prototype screw. In the case of buckwheat straw, the modification of the plasticizing system caused a decline in energy consumption of extrusion-cooking. Only when using the lowest process variables (20% moisture and 70 rpm), higher SME were recorded compared to the standard screw. In the work of Khor et al. [34], they studied the extrusion-cooking of grasses through fast and slow extrusion. They demonstrated that high-speed extrusion consumed 0.0867 ± 0.0021 kWh kg^−1^ grass, while the slow one required 0.140 ± 0.009 kWh kg^−1^ grass, which is much less than compared to the tested straws. Table 2 contains equations showing the functions of matching the response surface model for the examined feature based on the variables used.

### 3.3. Water Absorption Index Results

The substrates used in agricultural biogas plants are exposed to various treatments aimed at facilitating the methane fermentation process. One of such techniques is extrusion-cooking, which destructs the lignocellulosic structures and, as a result, enhances water absorption [19]. The WAI value shows the extent to which a processed material absorbs water. It can have a significant impact on how raw material responds in a fermentation chamber. A higher WAI may affect the water solubility index (WSI). As a result, the input fed into a fermentation chamber will be better mixed and will settle to the bottom, thus preventing the formation of scum inside the chamber [31]. The WAI value for the control sample (not extruded) was 7.618 g g^−1^ for buckwheat and 7.730 g g^−1^ for rapeseed.

Figure 8 shows the effect of the moisture content in raw material and the rotational speed of a standard extruder screw on the water absorption index. The WAI for processed rapeseed straw (Figure 8A) at the screw speed of 70 rpm was recorded low with the shift of the moisture level from 20 to 25% and from 30 to 35%; in contrast, it increases along with the rise of this level from 25 to 30%. The opposite dependence was observed for the screw speed of 90 rpm. In the case of the screw speed of 110 rpm, along with the increasing moisture content, the WAI keeps falling until 30% of the moisture level, while at 35% moisture, it nearly doubles. The highest measurement of the WAI (10.092 g g^−1^) for extruded rapeseed straw was recorded at the screw speed of 110 rpm and at 35% moisture. On the other hand, the lowest WAI (4.261 g g^−1^) was reported for the screw speed of 70 rpm and a 30% level of raw material moisturization.

For the extrusion of buckwheat straw (Figure 8B), the top WAI (6.87 g g^−1^) was observed in a sample pretreated at 35% moisture and the screw speed of 70 rpm. The lowest value (4.64 g g^−1^) was reported in the extrudate processed with a moisture content of 25% and the extruder screw speed of 70 rpm. Similar results were obtained by Żelaziński [35], who studied the extrusion-cooking of maize kernels and buckwheat grains. The WAI results for buckwheat grains were 5.04 g g^−1^ and for maize kernels 4.66 g g^−1^.

Figure 9 shows the results of the measurement of the WAI of extruded rapeseed and buckwheat straws when a prototype extruder screw was used. For rapeseed straw (Figure 9A), extruded at the screw speed of 110 rpm and with the higher moisture content, the value of the WAI increased across the entire range of completed tests. At the screw speed of 70 rpm, the WAI increased along with the change in the moisture level from 20 to 25% and from 30 to 35%. The opposite dependence was reported for the extruder screw speed of 90 rpm. The top WAI (6.623 g g^−1^) was recorded for a sample processed at the screw speed of 90 rpm and at 30% moisture. The lowest measurement result (4.696 g g^−1^) was obtained for a sample processed at the moisture level of 20% and the screw speed of 110 rpm.

In the case of buckwheat straw (Figure 9B), a higher level of moisture caused measurement fluctuations. At the extruder screw speed of 70 rpm, an increase in the WAI was recorded when the initial moisture level increased from 20 to 25 and 30%; after the level reached 35%, the WAI began to decline. The opposite trend was observed when the screw speed of 90 rpm was applied. For the screw speed of 110 rpm, an increase in the WAI was noted after the moisture level was raised from 20 to 25% and from 30 to 35%. The lowest measurement (4.955 g g^−1^) was obtained for samples processed at the moisture level of 25% and the screw speed of 90 rpm. The highest measured value (6.948 g g^−1^) of the WAI was also seen for the same screw speed and 35% moisture.

When the prototype screw was used, in most cases, buckwheat straw showed an increase in the WAI compared to the processing with the use of the standard screw. The WAI dropped only in two trials (at 90 rpm and 25% MC and at 70 rpm and 35% MC). For rapeseed straw, the modification of the plasticizing system increased the WAI at 30% of the moisture content across the entire range of screw speeds; the same was reported for trials at 70 rpm and 25% moisture; as regards 35% moisture, the WAI grew at the screw speeds of 70 and 90 rpm. Table 3 contains equations showing the functions of matching the response surface model for the examined feature based on the variables used.

### 3.4. Water Solubility Index Results

Another vital parameter is the water solubility index (WSI). A higher WSI proves the destruction of lignocellulose structures in the extrusion-cooking process, which is also related to the higher biogas efficiency of the substrate. The WSI determines to what extent the processed material can dissolve in water, i.e., it shows the number of soluble particles after moisturizing. A more compact material absorbs water to a lesser degree, which makes the solubility index lower. Pretreatment causes the material to deconstruct into smaller particles, thus reducing the density of the structure, which, in turn, accelerates the process of methane fermentation [35]. The WSI value for the control sample (not extruded) was 2.951% for buckwheat and 6.143% for rapeseed.

When measuring the WSI of rapeseed straw extrudates obtained with the standard plasticizing system (Figure 10A), at the screw speed of 110 rpm, the WSI value decreased along with the rising moisture content from 20 to 25%. In the other tested cases, the value increased. At the extruder screw speed of 70 rpm, an increase in the WSI was noted when the moisture level grew from 20 to 25 and 30%; after a further increase from 30 to 35%, the WSI value began to drop. For the screw speed of 90 rpm, the change in the moisture content from 20 to 25 and from 30 to 35% caused an increase in the WSI value; after the same content was raised from 25 to 30%, a slight decrease in the WSI was noted. The highest WSI (11.942%) was recorded in the extrudates produced at the extruder screw speed of 110 rpm and 20% moisture. When the lowest level of moisture and the slowest screw speed were applied, the WSI returned the lowest value (3.568%).

In the case of buckwheat straw extrudates (Figure 10B), the highest WSI value (12.896%) was observed after processing at the level of 20% of moisture and the rotational speed of 110 rpm. The lowest WSI (3.565%) was recorded in a sample pretreated at the level of 35% of moisture and the application of the screw speed of 70 rpm. It was also confirmed that the highest moisture level was covered by the lowest WSI value across the entire range of extruder screw speeds. For the screw speeds of both 70 and 90 rpm, the top WSI value was obtained in samples with a moisture content of 25%. For the screw speed of 110 rpm, the change in the moisture content from 20 to 25 and from 30 to 35% led to a drop in the WSI value; after the MC was raised from 25 to 30%, an increase in the WSI was noted.

Figure 11 shows the results of measurements of the WSI of extrudates when the prototype extruder screw was applied. In the case of rapeseed straw (Figure 11A), it was noted that for samples extruded at the screw speed of 90 and 110 rpm, raising the moisture level from 20 to 25% resulted in an increase in the WSI; when the moisture value was increased from 25 to 30 and 35%, the reported WSI was lower. At the screw speed of 70 rpm, a lower WSI value compared to the moisture levels given above was noted only for 35% moisture. The highest WSI (15.495%) was evaluated in samples processed at the extruder screw speed of 110 rpm and 25% moisture. The lowest solubility (4.262%) was obtained in a sample extruded at the screw speed of 110 rpm and at 35% moisture. In a similar study [33], extruded buckwheat grains showed a much higher WSI value (22.63%). Moreover, tested brown rice samples displayed an almost two times lower WSI than buckwheat grains (up to 14.32%) [32], which is similar to the straw results described in this paper.

In the case of buckwheat straw (Figure 11B), only at 35% of the moisture level, the samples produced at the rotational speed of the extruder screw of 110 rpm reached the lowest values; in the remaining cases, they showed the highest values in relation to the level of moisture. The opposite was observed in samples processed at the screw speed of 70 rpm, where only the level of 35% moisture coincided with the highest values; for all the remaining cases, the samples processed at this rpm returned the lowest WSI. The highest WSI (14.941%) was obtained in a sample processed at the moisture level of 30% and the screw speed of 110 rpm. The lowest WSI measurement (4.479%) was found in a sample extruded at the screw speed of 70 rpm and the level of moisture of 25%.

In the case of extruded buckwheat straw, the use of the modified plasticizing system (prototype) increased the WSI in samples processed at 30 and 35% moisture (across the entire range of the screw speeds applied) and at 20% moisture and the screw speeds of 90 and 110 rpm. As regards extruded rapeseed straw, the same trend was noted for samples processed with the use of 20% of moisture content (across the entire range of the screw speeds applied) and for samples extruded at 70 rpm and 30% moisture and 110 rpm and 25 and 30% moisture. Table 4 contains equations showing the functions of matching the response surface model for the examined feature based on the variables used.

### 3.5. Bulk Density of Extruded Straw

Another part of the research was aimed to determine the bulk density (BD) of the tested extrudates. The influence of process variables (raw material moisture, rotational speed of the extruder screw, and modification of the plasticizing system) on the BD of produced extrudates was measured. The use of the most preferred pretreatment method may help reduce the size of storage areas for bulk materials. The thermal processes occurring during the pretreatment make the raw material sterile and allow its storage for a longer period with no harm to quality. A higher level of BD may suggest a higher density of the material structure. It may also be true of WAI and WSI, thus having an impact on the efficiency of methane fermentation [19,36]. The BD value for the control sample (not extruded) was 210 kg m^−3^ for buckwheat and 105 kg m^−3^ for rapeseed.

For rapeseed straw processed with the standard plasticizing system (Figure 12A), the screw speed of 70 rpm and the rising moisture content increases the BD value, which falls again after the 30% moisture level is exceeded. On the other hand, at the extruder screw speeds of 90 and 110 rpm, the growing level of moisture raises the BD, which slightly decreases after the top moisture level has been reached. The highest BD obtained for the rapeseed straw extrudate (186 kg m^−3^) was recorded for samples prepared at the screw speed of 70 rpm and the moisture content of 25%. The lowest BD (68 kg m^−3^) was noted in samples obtained at the screw speed of 110 rpm and the moisture content of 20%.

As regards buckwheat straw extrudates (Figure 12B), the highest BD (252 kg m^−3^) was noted in samples pretreated at the moisture content of 20% and the extruder screw speed of 110 rpm. Along with the rising level of moisture, the BD gradually decreases across the entire range of rotational speeds. The lowest BD values (105 kg m^−3^) were recorded in samples obtained at the screw speeds of 90 and 110 rpm and 35% moisture.

During the measurements of the BD, higher values were recorded when buckwheat straw was processed using the prototype extruder screw (Figure 13B). For this raw material, the top BD values were recorded at 35% moisture. Only when the rotational speed of the extruder screw was set to 110 rpm, the value of the BD measurements increased along with the increase in the level of moisture. For the screw speeds of 70 and 90 rpm, the change in the moisture content from 20 to 25 and from 30 to 35% caused an increase in the BD value; after the same content was raised from 25 to 30%, a decrease in the BD was noted. The top BD (115 kg m^−3^) was recorded for a sample processed at the screw speed of 110 rpm and at 35% moisture. The lowest value (60 kg m^−3^) was noted in a sample obtained at the screw speed of 110 rpm and the moisture content of 20%.

In the case of rapeseed straw (Figure 13A), the highest BD values were also recorded for samples produced at 35% moisture. For samples obtained at the extruder screw speed of 90 rpm, a change to the moisture level increased the BD value. In samples extruded at the screw speeds of 70 and 110 rpm, the change in the moisture content from 20 to 25 and from 30 to 35% caused an increase in the BD value; after the MC was raised from 25 to 30%, a decrease in the BD was observed. The top values (87 kg m^−3^) were noted for samples extruded at 35% moisture and at the screw speeds of 70 and 110 rpm while the lowest (46 kg m^−3^) was observed in a sample obtained at the screw speed of 110 rpm and the moisture content of 30%.

As Kraszkiewicz et al. [37] reported, the BD of fuel pellets tested was 485.9 kg m^−3^ (rye straw and rape cake) and 521.4 kg m^−3^ (rye straw and soya bean cake), i.e., more than twice as high as in the case of the tested of the buckwheat straw and rapeseed straw extrudates described in this paper. The use of the modified plasticizing system reduced BD in both tested raw materials. Only in buckwheat straw processed at the screw speed of 110 rpm and at 35% moisture was a decrease in the BD noted, unlike with the samples processed using the prototype screw. Table 5 contains equations showing the functions of matching the response surface model for the examined feature based on the variables used.

### 3.6. Biogas Efficiency of the Extruded Straws

The extrusion-cooking process is among the raw material pretreatment methods used in biogas plants. This paper focuses on lignocellulosic raw materials. During the extrusion-cooking of these raw materials, lignin bonds are broken, and cellulose and hemicelluloses are released. The surface of active biomass grows, followed by its initial hydrolysis, which results in a higher biogas yield. The extrusion-cooking process should be preceded by proper preparation of the raw material used. The moisture content and the degree of fragmentation (grinding) are key factors. Finer grinding allows a more effective mixing of the raw material, which makes it more homogeneous. Proper selection of the moisture level significantly affects the extrusion-cooking process. Too low a level of moisture may cause the accumulation of raw material on the cylinder walls and the extruder screw. This, in turn, may lead to the overheating of the raw material and, consequently, interrupting of the process. Thus, the right moisture content of raw material enables its processing, regardless of the structure [19].

The efficiency of biogas production from the extruded rapeseed and buckwheat straws was studied at the last stage of the research. This study was aimed to perform the final comparison of the impact of modification of the plasticizing system on biogas efficiency and demonstrate that extrusion-cooking should be the preferred method of pretreatment of these raw materials if potentially used in agricultural biogas plants. The methane content in biogas in all the tested samples was at a similar level: for extruded rapeseed straw (Table 6), the control sample showed 58.59%, and the processed samples showed the range of 57.81–59.85%. It was observed that only in two cases (at 110 rpm and 20% MC and at 110 rpm and 35% MC), the methane content was lower than in the control sample (not pretreated).

In extruded buckwheat straw (Table 7), pretreatment reduced the methane content. In the same raw material, the methane content ranged from 52.22 to 53.20%; therefore, it was lower across the entire range of process variables than in the control sample (53.84%). The results for cumulative methane and biogas were compared in relation to fresh mass, dry mass, and dry organic mass. The application of extrusion-cooking as a pretreatment technique for rapeseed straws used as a source of biomass returned higher methane and biogas yields compared to those for extruded buckwheat straw. In the case of buckwheat straw, the lowest measurement values in relation to dry mass, fresh mass, and dry organic mass were reported for samples produced at the extruder screw speed of 90 rpm and the moisture level of 30%. At the same time, they returned the highest methane content. The highest measurement values were recorded for samples processed at the extruder screw speed of 110 rpm and at 25% moisture. In the case of extruded buckwheat straw, the highest measurement values were seen in samples processed at the screw speed of 70 rpm and at 35% moisture. The same rotational speed and 25% moisture content coincided with the lowest values for fresh mass and dry mass. In the case of dry organic mass, the lowest biogas value in a control sample was reported for the sample obtained at 20% of the moisture content and at the lowest extruder screw speed; in the case of methane, the lowest measurement value was recorded for the sample obtained at 25% of the moisture content and 90 rpm.

Generally, for fresh mass, the production of cumulative biogas ranged from 95.99% (when extruded at 90 rpm and 30% MC) to 110.30% (at 110 rpm and 25% MC) (in both cases for rapeseed straw), and of methane from 97.26% (buckwheat, at 70 rpm and 25% MC) to 111.57% (rapeseed, at 110 rpm and 25% MC). The higher biogas productivity for dry mass ranged from 95.85% (rapeseed, at 90 rpm and 30% MC) to 111.34% (buckwheat, at 70 rpm and 35% MC) and in the case of methane, from 96.97% (buckwheat, at 70 rpm and 25% MC) to 111.65% (rapeseed, at 110 rpm and 25% MC). In the case of dry organic mass, these values for biogas ranged from 95.43% (rapeseed, at 90 rpm and 30% MC) to 110.82% (buckwheat, at 70 rpm and 35% MC), and in the case of cumulative methane from 97.46% (rapeseed, at 90 rpm and 30% MC) to 111.74% (rapeseed, at 110 rpm and 25% MC).

Modification of the plasticizing system caused a decrease in the cumulative methane value (Table 8) compared with the raw material processed with no alterations to the plasticizing system (Table 6). In the case of extruded rapeseed straw processed after modification of the plasticizing system, as shown in Table 8, the methane content was lower than in a control sample (not extruded) across the entire range of process variables and ranged from 51.98% (when extruded at 70 rpm and 20% moisture and at 110 rpm and 30% moisture) to 54.34% (at 110 rpm and 35% moisture). Furthermore, the same process variables returned the lowest measured values over the entire range of cumulative gas and methane. The highest measurement values were seen in a sample extruded at the screw speed of 90 rpm and the level of moisture of 25%. As regards cumulative methane production in relation to dry mass and dry organic mass, the obtained measurements were lower than in the control sample across the entire range of process variables. The maximum results of biogas (400.64 m^3^ Mg^−1^) and methane yield (209.90 m^3^ Mg^−1^) from fresh mass were higher than in the silage obtained from pretreated maize straw. According to the author [38], the top values were 388.52 m^3^ Mg^−1^ of biogas and 205.68 m^3^ Mg^−1^ of methane.

In extruded buckwheat straw pretreated by modifying the plasticizing system, there was an increase in the production of biogas and methane compared with the control sample (not subjected to the extrusion process) across the entire range of tests (Table 9). As in the case of buckwheat straw extruded before the modification of the plasticizing system (Table 7), lower methane content values were recorded in the tests than in the control sample. The lowest values of cumulative biogas and methane were demonstrated in samples obtained at the screw speed of 90 rpm and 20% of the moisture content in relation to fresh mass, dry mass, and dry organic mass. The change in the moisture level of the raw material by 5% resulted in a significant increase in the yield of biogas and methane; top results were obtained in almost every case. Only in the case of fresh mass, the highest measured value of biogas yield was recorded for samples extruded at the top screw speed and 25% of the raw material moisture level. It was also observed that the highest measured values linked to extruder screw speeds were recorded for 25% of the moisture content across the entire range of applied speeds. For the speeds of 70 and 110 rpm, the lowest values were recorded at 35% of the moisture level. At the screw speed of 90 rpm, the lowest values were seen in samples with 20% of the moisture content.

In general, for the samples (rapeseed and buckwheat) processed after modification of the plasticizing system, and given the results for the control samples for fresh mass, the cumulative production of biogas ranged from 99.43 (rapeseed, pretreated at 110 rpm and 35% MC) to 123.24% (buckwheat processed at 110 rpm and 25% MC), and methane from 92.22 (rapeseed, at 110 rpm and 35%) to 115.98% (buckwheat, at 90 rpm and 25%). The cumulative production for dry mass ranged from 97.43 (rapeseed, at 110 rpm and 35%) to 116.77% (buckwheat, at 90 rpm and 25%) of biogas and from 90.36 (rapeseed, at 110 rpm and 35%) to 109.81% (buckwheat, at 90 rpm and 25%) of methane. As regards dry organic mass, the range was from 97.24 (rapeseed, at 110 rpm and 35%) to 116.69% (buckwheat, at 90 rpm and 25%) of biogas and from 90.18 (rapeseed, at 110 rpm and 35%) to 109.72% (buckwheat, at 90 rpm and 25%) of methane. The determination of biogas and methane yields, when calculated on fresh matter, dry organic matter or in total solids and volatile solids, is necessary for the comparison of energetic potential between substrates [38,39].

### 3.7. Principal Component Analysis (PCA) Results

The obtained results were used for the principal component analysis (PCA) before and after modification of the extruder plasticizing system and objects were located within the first two components. Presentation of all the results in the form of a graph (Figure 14) offers a broader view and the possibility of uncovering links between the process variables and obtained results.

Approximately 71% (Figure 14A) of all the variance in data, if screw without modification was applied in the experiment, was explained by the first two principal components PC1 (56.20%) and PC2 (15.07%). PC1 was positively correlated with BD and strongly negatively correlated with FM-CCH4, FM-CB, TS-CCH4, TS-CB, VS-CCH4 and VS-CB. PC2 was positively correlated with Q and negatively correlated with SME and WAI. After placing the samples in the space of the first two components (Figure 14B,D), results differ from each other depending on the raw material. Various correlations between the tested parameters were also observed. Q was strongly negatively correlated with SME. BD was negatively correlated with FM-CCH4, TS-CCH4 and (VS-CCH4). Strong positive correlations were observed between FM-CCH4, FM-CB, TS-CCH4, TS-CB, VS-CCH4 and VS-CB. PC1 (Figure 14C) was positively correlated with BD, Q, and WAI and negatively correlated with SME, FM-CCH4, TS-CCH4 VS-CCH4, FM-CB, TS-CB, and VS-CB. PC2 was positively correlated with FM-CCH4 and TS-CCH and negatively correlated with SME and WSI. Based on the conducted analysis, relations between the individual parameters were observed. SME was negatively correlated with Q and BD and positively correlated with WSI, FM-CB, VS-CCH4 and VS-CB. The measurement results are strongly oriented (rapeseed straw on the left and buckwheat straw on the right side of Figure 14B,D), which may suggest the nature of the obtained results.

## 4. Conclusions

The following conclusions were drawn based on the conducted tests and the analysis of the results. The results in individual tests differed depending on the raw material used. Not only may this be attributed to the structure of the tested raw material, but it may also suggest the need to select appropriate process parameters for a specific material. In the case of buckwheat straw, the use of a prototype extruder screw increased the efficiency of the extrusion-cooking process and reduced its energy consumption. In the case of rapeseed straw, modification of the plasticizing system raised the energy requirement and reduced the efficiency of the process compared to when a standard screw was used. The opposite trend was reported only when 35% of the moisture content was applied.

When the prototype screw was used, in most cases, extruded buckwheat straw showed an increase in the WAI compared to the processing with the use of the standard screw. For rapeseed straw, the WAI index was noted to depend on the moisture level of the raw material. In the case of 20 and 25% of the moisture content, a higher WAI was observed in samples processed with the standard screw; and when moisture was raised, a higher WAI was noted in samples processed with the prototype screw.

In the case of extruded buckwheat straw, the use of the modified plasticizing system (prototype) increased the WSI in samples processed at 30 and 35% moisture (across the entire range of the screw speeds applied). In extruded rapeseed straw, this trend was observed for samples processed at a moisture level of 20%.

The use of the modified plasticizing system reduced BD in both tested raw materials. Only in buckwheat straw processed at the screw speed of 110 rpm and at 35% moisture was a decrease in the BD noted, unlike with the samples processed using the prototype screw.

In extruded buckwheat straw (obtained in the modified plasticizing system), there was an increase in the production of biogas and methane compared with the control sample (not subjected to the extrusion process) across the entire range of tests. For samples processed with the standard screw, the best yield of biogas across the entire range of tests was obtained for the sample processed at the screw speed of 70 rpm and 35% of the moisture content. Rapeseed straw processed with the prototype screw produced the highest biogas yield at the screw speed of 90 rpm applied for pretreatment. When testing rapeseed straw processed with the use of the standard screw, the top measured values were reported in samples processed at the maximum rotational speed of the extruder screw. In both cases, the best results were found in samples processed with a moisture level of 25%.

In future research attempts, more attention should be paid to the proper selection of process variables for specific raw materials. It would be reasonable to carry out a comparative analysis using various extruder plastification unit configurations (L/D) or screw lengths, as well as measuring the influence of modification of the plasticizing system on the extrusion-cooking process stability and on methane efficiency.

## Figures and Tables

**Figure 1 materials-15-05039-f001:**
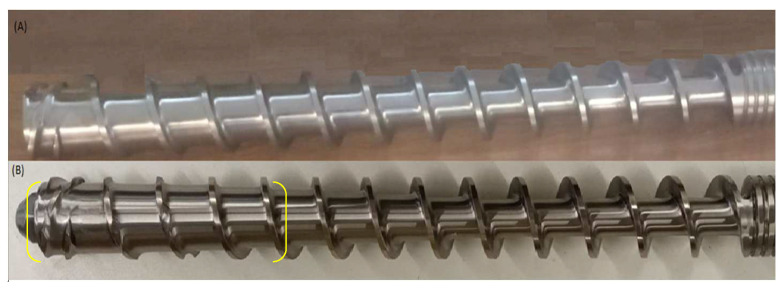
Part of the plasticizing system: (**A**) standard screw; (**B**) prototype (modified) screw.

**Figure 2 materials-15-05039-f002:**
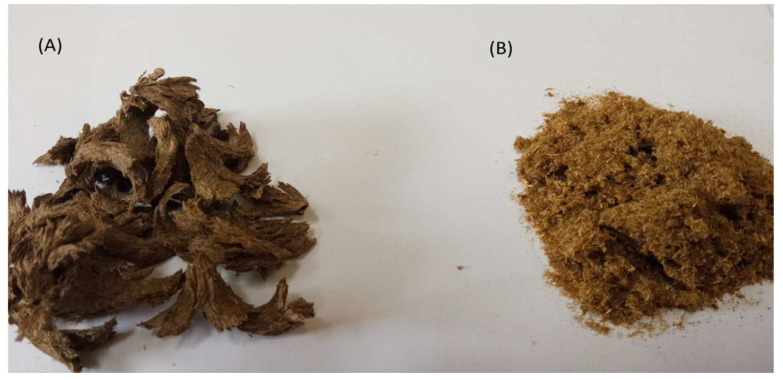
View of obtained sample extrudates: (**A**) from buckwheat straw; (**B**) from rapeseed straw.

**Figure 3 materials-15-05039-f003:**
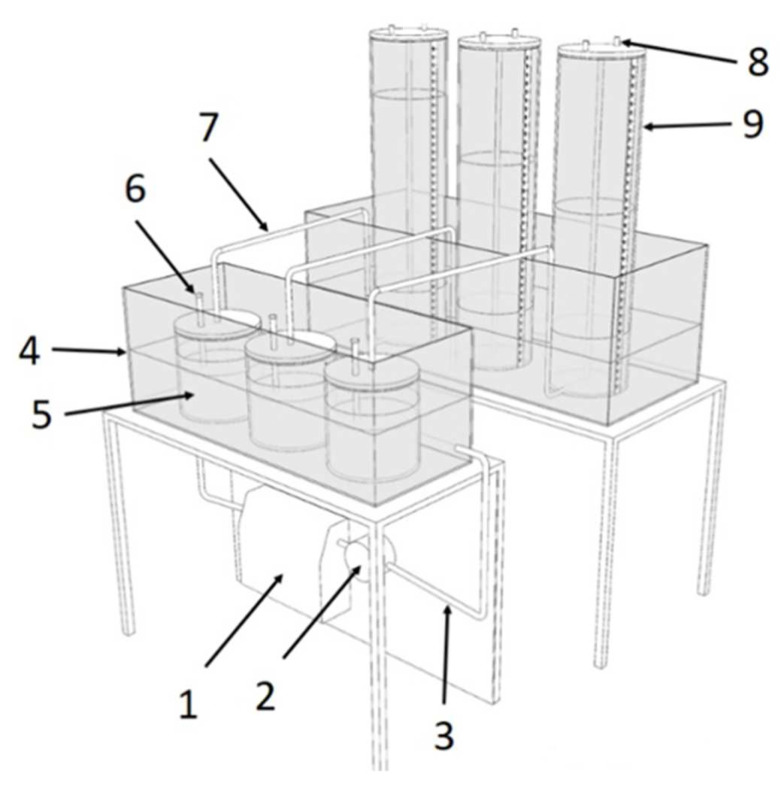
A design of a biofermentor for biogas production research (3-chamber section) [26]: 1—water heater with temperature regulator, 2—water pump, 3—insulated conductors of calefaction liquid, 4—water coat, 5—biofermentor with the charge capacity of 2 dm^3^, 6—sampling tubes, 7—biogas transporting tube, 8—gas sampling valve, 9—biogas volume-scale reservoir.

**Figure 4 materials-15-05039-f004:**
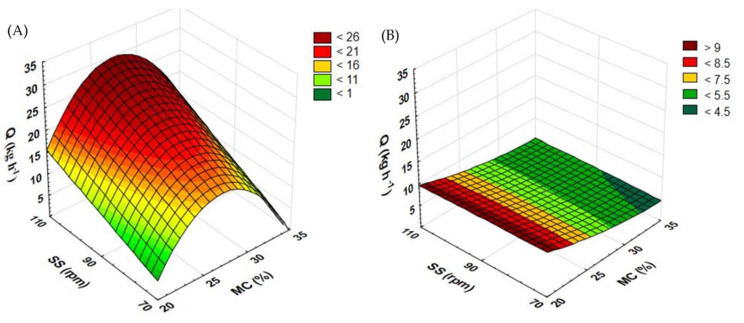
Efficiency of the extrusion-cooking of rapeseed straw (**A**) and buckwheat straw (**B**) before modification of the plasticizing system: MC—moisture content; SS—screw speed; Q—efficiency.

**Figure 5 materials-15-05039-f005:**
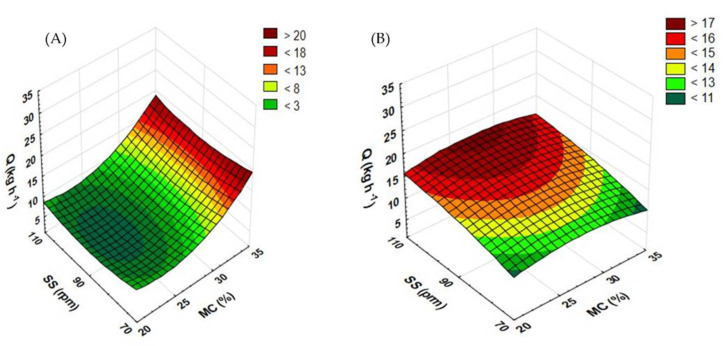
Efficiency of the extrusion-cooking of rapeseed straw (**A**) and buckwheat straw (**B**) after modification of the plasticizing system: MC—moisture content; SS—screw speed; Q—efficiency.

**Figure 6 materials-15-05039-f006:**
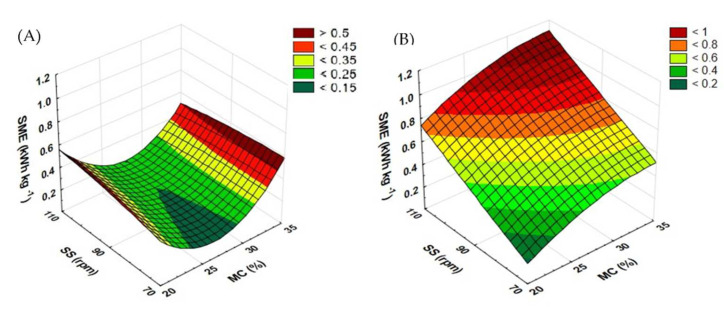
Specific mechanical energy of the extrusion-cooking of rapeseed straw (**A**) and buckwheat straw (**B**) before modification of the plasticizing system: MC—moisture content; SS—screw speed; SME—specific mechanical energy.

**Figure 7 materials-15-05039-f007:**
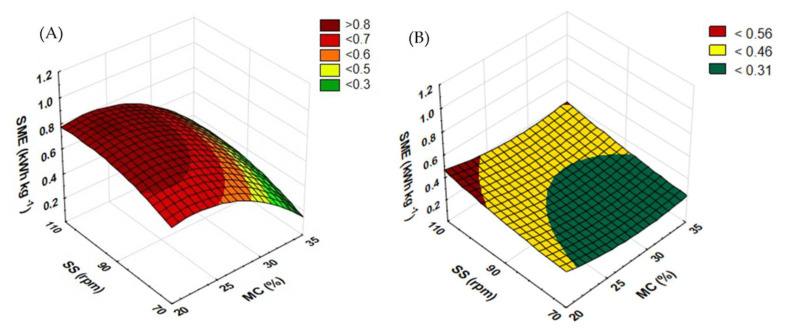
Specific mechanical energy of the extrusion-cooking of rapeseed straw (**A**) and buckwheat straw (**B**) after modification of the plasticizing system: MC—moisture content; SS—screw speed; SME—specific mechanical energy.

**Figure 8 materials-15-05039-f008:**
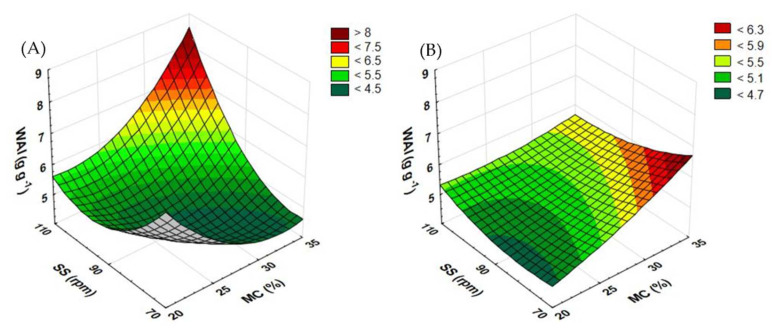
The results of WAI measurement for extruded rapeseed straw (**A**) and buckwheat straw (**B**) before modification of the plasticizing system: MC—moisture content; SS—screw speed; WAI—water absorption index.

**Figure 9 materials-15-05039-f009:**
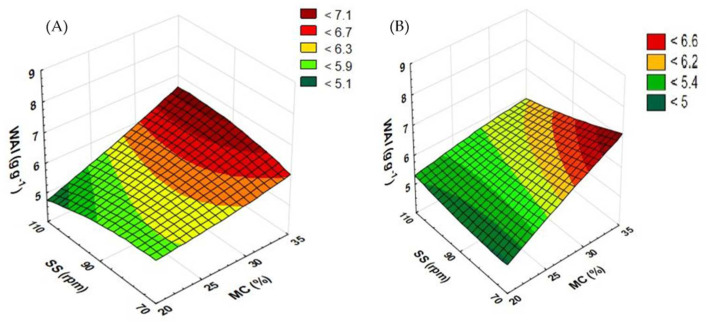
The result of WAI measurement for extruded rapeseed straw (**A**) and buckwheat straw (**B**) after modification of the plasticizing system: MC—moisture content; SS—screw speed; WAI—water absorption index.

**Figure 10 materials-15-05039-f010:**
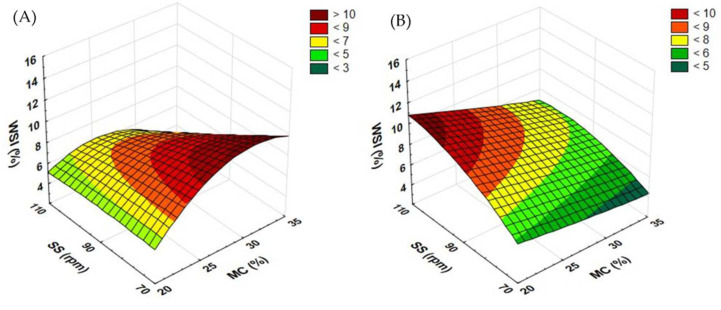
Results of the WSI of extrudates obtained from rapeseed straw (**A**) and buckwheat straw (**B**) before modification of the plasticizing system: MC—moisture content; SS—screw speed; WSI—water solubility index.

**Figure 11 materials-15-05039-f011:**
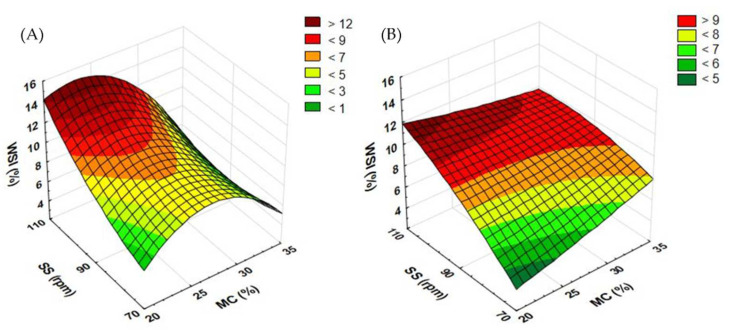
Results of the WSI of extrudates obtained from rapeseed straw (**A**) and buckwheat straw (**B**) after modification of the plasticizing system: MC—moisture content; SS—screw speed; WSI—water solubility index.

**Figure 12 materials-15-05039-f012:**
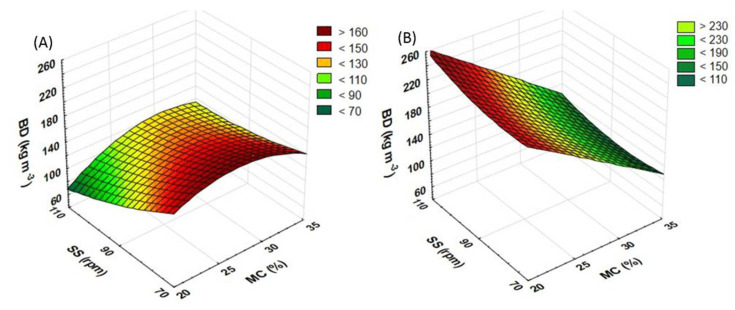
The result of BD measurement for extruded rapeseed straw (**A**) and buckwheat straw (**B**) before modification of the plasticizing system: MC—moisture content; SS—screw speed; BD—bulk density.

**Figure 13 materials-15-05039-f013:**
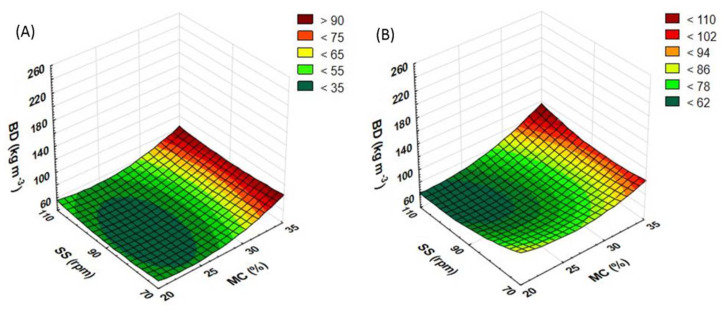
The result of BD measurement for extruded rapeseed straw (**A**) and buckwheat straw (**B**) after modification of the plasticizing system: MC—moisture content; SS—screw speed; BD—bulk density.

**Figure 14 materials-15-05039-f014:**
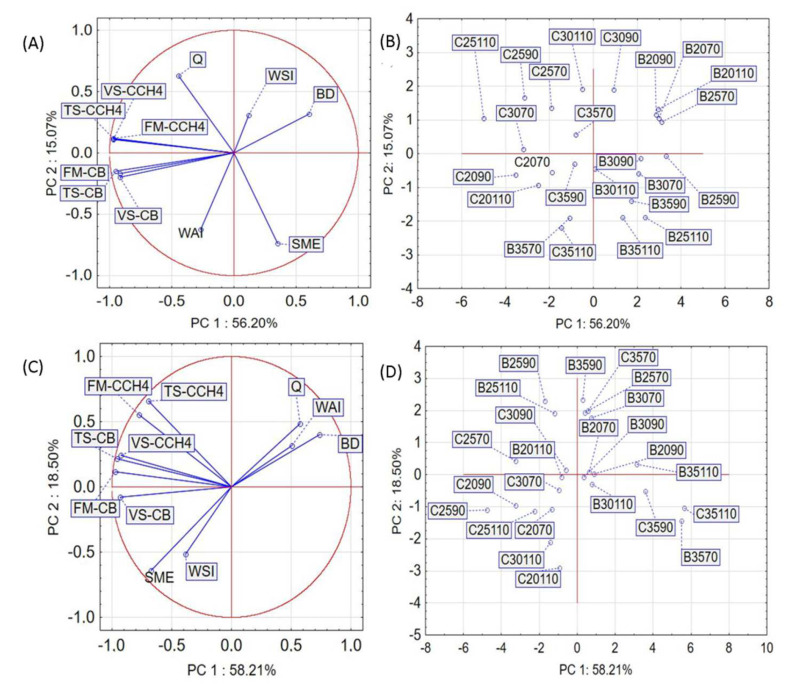
Principal component analysis (PCA) before (**A**) and after modification (**C**) and objects within the first two major components (**B**—before modification) (**D**—after modification): B—buckwheat; C—rapeseed; Q—efficiency; SME—specific mechanical energy; WAI—water absorption index; WSI—water solubility index; BD—bulk density; FM-CCH4—cumulative methane production (fresh matter); FM-CB—cumulative biogas production (fresh matter); TS-CCH4—cumulative methane production (total solids); TS-CB—cumulative biogas production (total solids); VS-CCH4—cumulative methane production (volatile solids); VS-CB—cumulative biogas production (volatile solids).

**Table 1 materials-15-05039-t001:** Models of response surface approximation for the efficiency of extrusion-cooking of rapeseed and buckwheat straws depending on the process variables.

Raw Material	Type of Screw	Response Surface Approximation Model (Surface Polynomial)
Rapeseed	a	Q (kg h^−1^) = −181.1829 + 13.6369SS + 0.1061MC − 0.2631SS^2^ + 0.007SSMC + 0.0002MC^2^
Buckwheat	a	Q (kg h^−1^) = 29.4703 − 1.4195SS + 0.0148MC + 0.0179SS^2^ + 0.0013SSMC − 0.0003MC^2^
Rapeseed	b	Q (kg h^−1^) = 84.042 − 3.6168SS − 0.7822MC + 0.0783SS^2^ − 0.0001SSMC + 0.0043MC^2^
Buckwheat	b	Q (kg h^−1^) = −33.8327 + 1.6877SS + 0.4532MC − 0.032SS^2^ + 0.0007SSMC − 0.0019MC^2^

a—before modification of plasticizing system; b—after modification of plasticizing system; Q—efficiency (kg h^−1^); SS—screw speed (rpm); MC—moisture (%).

**Table 2 materials-15-05039-t002:** Models of response surface approximation for the energy consumption of extrusion-cooking of rapeseed and buckwheat straws depending on the process variables.

Raw Material	Type of Screw	Response Surface Approximation Model (Surface Polynomial)
Rapeseed	a	SME (kWh kg^−1^) = 2.9961 − 0.2806SS + 0.021MC + 0.0059SS^2^ − 0.0004SSMC − 4.5625E − 5MC^2^
Buckwheat	a	SME (kWh kg^−1^) = −2.0726 + 0.0958SS + 0.0109MC − 0.0014SS^2^ + 5.3E − 5SSMC + 1.1875E − 5MC^2^
Rapeseed	b	SME (kWh kg^−1^) = −2.4543 + 0.1298SS + 0.035MC − 0.003SS^2^ + 5.2608E − 5SSMC − 0.0002MC^2^
Buckwheat	b	SME (kWh kg^−1^) = 1.348 − 0.0685SS − 0.0051MC + 0.0011SS^2^ + 2.8604E − 5SSMC + 4.5111E − 5MC^2^

a—before modification of plasticizing system; b—after modification of plasticizing system; SME—special mechanical energy (kWg kg^−1^); SS—screw speed (rpm); MC—moisture (%).

**Table 3 materials-15-05039-t003:** Models of response surface approximation for water absorption of extruded rapeseed and buckwheat straws depending on the process variables.

Raw Material	Type of Screw	Response Surface Approximation Model (Surface Polynomial)
Rapeseed	a	WAI (g g^−1^) = 58.0325 − 1.6762SS − 0.7154MC + 0.0161SS^2^ + 0.009SSMC + 0.0028MC^2^
Buckwheat	a	WAI (g g^−1^) = 4.79 + 0.1805SS − 0.0665MC + 0.0027SS^2^ − 0.0028SSMC + 0.0008MC^2^
Rapeseed	b	WAI (g g^−1^) = 6.1705 − 0.1736SS + 0.0212MC + 0.0011SS^2^ + 0.0021SSMC − 0.0005MC^2^
Buckwheat	b	WAI (g g^−1^) = −1.2092 + 0.3725SS + 0.0233MC − 0.001SS^2^ − 0.0025SSMC + 0.0002MC^2^

a—before modification of plasticizing system; b—after modification of plasticizing system; WAI—water absorption index (g g^−1^); SS—screw speed (rpm); MC—moisture (%).

**Table 4 materials-15-05039-t004:** Models of response surface approximation for the water solubility index of extruded rapeseed and buckwheat straws depending on the process variables.

Raw Material	Type of Screw	Response Surface Approximation Model (Surface Polynomial)
Rapeseed	a	WSI (%) = −51.7629 + 3.5307SS + 0.3303MC − 0.0433SS^2^ − 0.0121SSMC − 0.0005MC^2^
Buckwheat	a	WSI (%) = −24.2204 + 0.0938SS + 0.6451MC + 0.002SS^2^ − 0.0043SSMC − 0.0024MC^2^
Rapeseed	b	WSI (%) = −53.831 + 4.7531SS − 0.025MC − 0.0702SS^2^ − 0.0136SSMC + 0.0028MC^2^
Buckwheat	b	WSI (%) = −39.6139 + 0.7949SS + 0.6947MC + 0.0031SS^2^ − 0.0101SSMC − 0.0017MC^2^

a—before modification of plasticizing system; b—after modification of plasticizing system; WSI—water solubility index (%); SS—screw speed (rpm); MC—moisture (%).

**Table 5 materials-15-05039-t005:** Models of response surface approximation for the bulk density of extruded rapeseed and buckwheat straws depending on the process variables.

Raw Material	Type of Screw	Response Surface Approximation Model (Surface Polynomial)
Rapeseed	a	BD (kg m^−3^) = 235.1062 + 18.04SS − 6.7463MC − 0.45SS^2^ + 0.0885SSMC + 0.0166MC^2^
Buckwheat	a	BD (kg m^−3^) = 497.3625 − 5.62SS − 3.44MC − 0.01SS^2^ − 0.029SSMC + 0.0269MC^2^
Rapeseed	b	BD (kg m^−3^) = 286.3875 − 12.53SS − 2.105MC + 0.2767SS^2^ − 0.008SSMC + 0.0131MC^2^
Buckwheat	b	BD (kg m^−3^) = 601.6708 − 18.7767SS − 6.645MC + 0.27SS^2^ + 0.068SSMC + 0.0256MC^2^

a—before modification of plasticizing system; b—after modification of plasticizing system; BD—bulk density (kg m^−3^); SS—screw speed (rpm); MC—moisture (%).

**Table 6 materials-15-05039-t006:** Biogas efficiency of extruded rapeseed straw (before modification of the extruder screw; ±SD).

Material Type	Screw Speed (rpm)	Moisture Content (%)	Methane Content (%)	Cumulative Production (m^3^ per Mg Fresh Mass)		Cumulative Production (m^3^ per Mg Dry Mass)		Cumulative Production (m^3^ per Mg Dry Organic Mass)	
				Biogas	Methane	Biogas	Methane	Biogas	Methane
Control			58.59 ± 1.28	349.98 ± 1.20	205.07 ± 1.27	385.93 ± 1.45	226.13 ± 1.28	420.30 ± 1.32	246.28 ± 1.64
Extruded rapeseed	70	20	59.71 ± 1.45	362.63 ± 1.26	216.51 ± 1.49	401.64 ± 1.32	239.80 ± 1.45	435.85 ± 1.68	260.22 ± 1.76
		25	59.26 ± 1.35	363.84 ± 1.25	215.61 ± 1.16	404.25 ± 1.46	239.56 ± 1.27	441.72 ± 1.34	261.76 ± 1.35
		30	59.28 ± 1.25	381.00 ± 1.64	225.87 ± 1.83	419.20 ± 1.25	248.52 ± 1.23	455.25 ± 1.34	269.89 ± 1.46
		35	58.62 ± 1.23	356.07 ± 1.25	208.73 ± 1.24	397.17 ± 1.64	232.81 ± 1.25	432.73 ± 1.26	253.67 ± 1.39
	90	20	58.80 ± 1.53	380.51 ± 1.43	223.76 ± 1.27	420.41 ± 1.49	247.22 ± 1.45	457.16 ± 1.28	268.83 ± 1.69
		25	58.83 ± 1.49	373.98 ± 1.53	220.01 ± 1.87	412.34 ± 1.46	242.87 ± 1.45	449.40 ± 1.41	264.70 ± 1.02
		30	59.85 ± 1.87	335.95 ± 1.32	201.06 ± 1.01	369.92 ± 1.12	221.39 ± 1.21	401.08 ± 1.05	240.03 ± 1.72
		35	58.79 ± 1.46	354.44 ± 1.79	208.37 ± 1.64	399.00 ± 1.43	234.56 ± 1.25	432.19 ± 1.34	254.07 ± 1.27
	110	20	57.95 ± 1.21	375.61 ± 1.13	217.66 ± 1.54	408.00 ± 1.54	236.43 ± 1.21	441.89 ± 1.29	256.07 ± 1.09
		25	59.27 ± 1.13	386.04 ± 1.54	228.79 ± 1.61	425.98 ± 1.73	252.47 ± 1.46	464.35 ± 1.87	275.20 ± 0.97
		30	59.59 ± 0.54	344.41 ± 1.28	205.23 ± 1.45	382.74 ± 1.38	227.83 ± 1.28	415.53 ± 0.99	247.35 ± 1.25
		35	57.81 ± 1.21	357.10 ± 1.44	206.44 ± 1.75	397.48 ± 1.64	229.79 ± 1.07	431.03 ± 1.36	249.18 ± 1.41

**Table 7 materials-15-05039-t007:** Biogas efficiency of extruded buckwheat straw (before modification of the extruder screw; ±SD).

Material Type	Screw Speed (rpm)	Moisture Content (%)	Methane Content (%)	Cumulative Production (m^3^ per Mg Fresh Mass)		Cumulative Production (m^3^ per Mg Dry Mass)		Cumulative Production (m^3^ per Mg Dry Organic Mass)	
				Biogas	Methane	Biogas	Methane	Biogas	Methane
Control			53.84 ± 0.46	336.37 ± 1.12	181.09 ± 1.27	366.68 ± 1.64	197.53 ± 1.23	405.77 ± 1.24	218.46 ± 0.97
Extruded buckwheat	70	20	52.82 ± 1.11	340.01 ± 0.91	179.60 ± 1.46	370.01 ± 1.47	195.44 ± 1.45	403.89 ± 1.11	213.34 ± 1.32
		25	52.70 ± 0.99	334.23 ± 1.28	176.13 ± 1.25	363.49 ± 1.57	191.55 ± 1.31	406.50 ± 1.40	214.21 ± 1.54
		30	52.93 ± 1.01	340.19 ± 1.19	180.05 ± 1.68	371.80 ± 1.39	196.78 ± 1.42	413.04 ± 1.25	218.60 ± 1.05
		35	52.69 ± 0.87	370.47 ± 1.52	195.19 ± 1.64	408.26 ± 1.93	215.09 ± 0.79	449.68 ± 1.52	236.92 ± 1.46
	90	20	52.22 ± 1.54	341.94 ± 1.02	178.55 ± 1.47	370.20 ± 1.87	193.30 ± 1.62	410.14 ± 1.33	214.16 ± 1.37
		25	52.79 ± 1.21	335.63 ± 1.64	177.18 ± 1.36	365.91 ± 1.25	193.03 ± 1.69	404.38 ± 1.22	213.33 ± 1.25
		30	52.65 ± 1.64	343.75 ± 1.34	180.99 ± 1.20	377.19 ± 1.58	198.60 ± 1.31	413.80 ± 1.43	217.88 ± 1.47
		35	53.20 ± 1.54	345.86 ± 1.02	184.00 ± 1.34	377.85 ± 1.76	201.02 ± 1.01	417.76 ± 1.41	222.25 ± 1.11
	110	20	52.76 ± 1.24	342.19 ± 1.37	180.54 ± 0.90	373.04 ± 1.21	196.81 ± 1.37	407.86 ± 1.42	215.19 ± 1.07
		25	52.46 ± 1.24	349.57 ± 1.07	183.38 ± 1.21	381.91 ± 1.33	200.35 ± 1.12	420.50 ± 1.31	220.59 ± 1.41
		30	52.57 ± 0.93	365.45 ± 1.14	192.11 ± 1.27	404.35 ± 1.36	212.64 ± 1.07	440.58 ± 1.02	231.69 ± 1.12
		35	52.84 ± 0.98	351.32 ± 1.23	185.63 ± 1.49	383.83 ± 1.91	202.80 ± 1.75	424.52 ± 1.26	224.30 ± 1.41

**Table 8 materials-15-05039-t008:** Biogas efficiency of extruded rapeseed straw (after modification of the extruder screw; ±SD).

Material Type	Screw Speed (rpm)	Moisture Content (%)	Methane Content (%)	Cumulative Production (m^3^ per Mg Fresh Mass)		Cumulative Production (m^3^ per Mg Dry Mass)		Cumulative Production (m^3^ per Mg Dry Organic Mass)	
				Biogas	Methane	Biogas	Methane	Biogas	Methane
Control			58.59 ± 2.01	349.98 ± 1.93	205.07 ± 1.51	385.93 ± 1.24	226.13 ± 1.87	420.30 ± 1.22	246.28 ± 1.82
Extruded rapeseed	70	20	51.98 ± 1.45	388.28 ± 0.32	201.85 ± 1.54	412.38 ± 1.38	214.56 ± 1.45	456.27 ± 2.14	237.19 ± 1.95
		25	52.64 ± 2.21	396.76 ± 1.53	208.84 ± 1.73	424.33 ± 1.42	223.36 ± 1.29	466.19 ± 0.87	245.39 ± 1.29
		30	52.30 ± 1.37	386.75 ± 1.43	202.27 ± 1.64	414.24 ± 1.87	216.64 ± 1.43	450.29 ± 0.99	235.50 ± 1.01
		35	53.78 ± 1.75	380.95 ± 1.10	204.89 ± 1.03	408.68 ± 1.91	219.86 ± 1.34	442.86 ± 0.67	238.19 ± 2.03
	90	20	52.14 ± 1.64	398.12 ± 1.86	207.57 ± 1.94	420.95 ± 1.72	219.47 ± 1.29	462.41 ± 1.34	241.08 ± 1.97
		25	52.39 ± 1.34	400.64 ± 2.01	209.90 ± 2.15	428.38 ± 1.64	224.43 ± 1.46	468.41 ± 1.76	245.40 ± 1.01
		30	52.42 ± 1.85	386.93 ± 1.12	202.83 ± 1.73	413.46 ± 1.45	216.74 ± 1.64	451.32 ± 1.73	236.58 ± 1.30
		35	53.80 ± 1.23	362.96 ± 1.74	195.29 ± 1.02	388.05 ± 0.64	208.79 ± 1.54	418.78 ± 1.60	225.32 ± 1.21
	110	20	52.31 ± 2.21	382.55 ± 1.21	200.10 ± 2.37	404.90 ± 1.34	211.79 ± 1.64	444.18 ± 1.39	232.33 ± 1.42
		25	53.38 ± 1.82	386.11 ± 1.38	206.11 ± 1.94	413.39 ± 1.76	220.04 ± 1.09	453.54 ± 1.65	241.42 ± 1.38
		30	51.98 ± 1.66	388.53 ± 1.46	201.96 ± 2.01	413.15 ± 1.93	214.76 ± 1.43	448.13 ± 1.87	232.94 ± 1.15
		35	54.34 ± 1.64	347.99 ± 1.07	189.11 ± 1.94	376.01 ± 1.47	204.34 ± 1.67	408.69 ± 1.66	222.10 ± 1.06

**Table 9 materials-15-05039-t009:** Biogas efficiency of extruded buckwheat straw (after modification of the extruder screw; ±SD).

Material Type	Screw Speed (rpm)	Moisture Content (%)	Methane Content (%)	Cumulative Production (m^3^ per Mg Fresh Mass)		Cumulative Production (m^3^ per Mg Dry Mass)		Cumulative Production (m^3^ per Mg Dry Organic Mass)	
				Biogas	Methane	Biogas	Methane	Biogas	Methane
Control			57.38 ± 1.64	319.21 ± 1.70	183.18 ± 2.01	361.20 ± 1.16	207.24 ± 1.97	380.30 ± 1.12	218.23 ± 1.77
Extruded buckwheat	70	20	54.59 ± 1.81	375.19 ± 1.65	204.81 ± 1.63	400.80 ± 1.11	218.79 ± 1.64	423.39 ± 1.15	231.12 ± 1.37
		25	54.51 ± 1.26	380.45 ± 1.95	207.38 ± 1.34	408.35 ± 1.15	222.59 ± 2.34	431.07 ± 1.66	234.98 ± 1.94
		30	54.85 ± 1.92	377.98 ± 2.12	207.33 ± 1.62	405.14 ± 1.28	222.22 ± 1.13	426.46 ± 1.91	233.92 ± 1.35
		35	55.68 ± 1.64	346.65 ± 1.09	193.02 ± 1.15	372.41 ± 1.82	207.50 ± 1.37	393.79 ± 2.69	219.27 ± 1.37
	90	20	54.54 ± 1.58	375.55 ± 1.18	204.83 ± 1.55	398.85 ± 1.94	217.54 ± 1.45	421.60 ± 1.85	229.94 ± 1.28
		25	54.83 ± 1.86	387.50 ± 1.88	212.45 ± 1.73	421.78 ± 2.07	227.57 ± 1.99	443.78 ± 2.07	239.44 ± 1.98
		30	54.62 ± 1.64	377.28 ± 1.72	206.06 ± 1.85	401.96 ± 1.39	219.54 ± 1.92	423.93 ± 2.07	231.54 ± 1.43
		35	54.93 ± 1.11	381.38 ± 1.36	209.49 ± 1.29	409.66 ± 1.85	225.02 ± 1.46	430.96 ± 1.64	236.72 ± 1.97
	110	20	54.25 ± 2.33	383.77 ± 1.55	208.17 ± 1.15	408.25 ± 1.88	221.45 ± 1.66	431.32 ± 1.61	233.97 ± 1.52
		25	53.52 ± 1.99	393.38 ± 1.37	210.53 ± 1.37	419.34 ± 1.23	224.43 ± 1.60	441.80 ± 1.96	236.45 ± 1.94
		30	53.17 ± 1.82	380.54 ± 1.10	202.32 ± 2.07	409.20 ± 1.92	216.73 ± 1.99	430.52 ± 1.23	228.02 ± 2.02
		35	55.48 ± 1.73	361.02 ± 1.94	200.31 ± 1.25	389.36 ± 2.37	216.03 ± 1.65	410.71 ± 2.00	227.87 ± 1.28

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
