# Peer review of "Influence of Modification of the Plasticizing System on the Extrusion-Cooking Process and Selected Physicochemical Properties of Rapeseed and Buckwheat Straws"

_materials, 2022, doi:10.3390/ma15145039_

Round 1
Reviewer 1 Report
This paper aimed to analyse the effect of modification of the plasticizing system of a single-screw extruder on selected physicochemical properties of rapeseed straw and buckwheat straw. This paper is well-written and provides some interesting information. However, minor revision should be made before it is published in the Journal.
The comments are as follows:
1. The authors are encouraged to revise the abstract and improve it by inserting some results.
2. The introduction section is too long. I suggest shortening it from the line 28 to 95.
3. Are really necessary the citations of 6 articles of the corresponding author dr. Tomasz Oniszczuk?
Author Response
This paper aimed to analyse the effect of modification of the plasticizing system of a single-screw extruder on selected physicochemical properties of rapeseed straw and buckwheat straw. This paper is well-written and provides some interesting information. However, minor revision should be made before it is published in the Journal.
The comments are as follows:
- The authors are encouraged to revise the abstract and improve it by inserting some results.
Author’s reply: Thank you for the suggestion. The following information has been added: It has been proved that the modification of the plasticizing system had a significant impact on the course of the process and the tested physicochemical properties. An important factor confirming the correctness of the modification is the increase in biogas efficiency. After modification the highest yield of cumulative biogas from fresh mass was 12.94% higher than in the sample processed before modification.
- The introduction section is too long. I suggest shortening it from the line 28 to 95.
Author’s reply: We agree with the Reviewer. The Introduction section has been reduced accordingly.
- Are really necessary the citations of 6 articles of the corresponding author dr. Tomasz Oniszczuk?
Author’s reply: According to the Reviewer’s suggestion number of prof. Tomasz Oniszczuk citations has been reduced.
Reviewer 2 Report
Title: must be revised. Its extremely long.
Why do you have a full stop in your title?
Abstract: must be revised. What are the main results and conclusions of your study?
Keywords: Must be different from the title to increase the visibility of your paper.
Introduction:
You didn't connect the paragraph on line 66 with the previous one. You start talking about biogas out of the blue.
Line 74: remove extra space
Line 82: remove extra space
Line 84: Rapeseed should be capital letter
What are the objectives of your study? You must add a paragraph at the end of the introduction with this information. This is the basics of any paper.
Materials and methods:
Line 129: How did you measure the moisture content? Which equipment was used?
Results and discussion:
Biogas and biomethane results should be presented in a Figure, not in a table. You don't need to show the results in 3 different units. Pick one and make a figure out of it. You will also be able to combine the results of extruded rapeseed straw, extruded rapeseed straw (after modification of the extruder screw), and extruded buckwheat straw (after modification of the extruder screw) in a single figure to allow comparison.
You must add standard deviations to all your results in the tables and in the figures your will create.
Author Response
Title: must be revised. Its extremely long.
Author’s reply: The title of the work has been shortened.
Was: Influence of Modification of the Plasticizing System of a Single-screw Extruder on the Extrusion-cooking Process and Selected Physicochemical Properties of Rapeseed straw and Buckwheat Straw
Now it is: Influence of the Extrusion-cooking Process Modification on Selected Physicochemical Properties of Rapeseed and Buckwheat Straws
Why do you have a full stop in your title?
Author’s reply: The dot has been removed.
Abstract: must be revised. What are the main results and conclusions of your study?
Author’s reply: Thank you for the suggestion. The following information has been added: It has been proved that the modification of the plasticizing system had a significant impact on the course of the process and the tested physicochemical properties. An important factor confirming the correctness of the modification is the increase in biogas efficiency. After modification the highest yield of cumulative biogas from fresh mass was 12.94% higher than in the sample processed before modification.
Keywords: Must be different from the title to increase the visibility of your paper.
Author’s reply: Thank you for your suggestion. As proposed by the Reviewer, the keywords have been changed.
Was: extrusion-cooking; SME; biogas; methane; rapeseed straw; buckwheat straw
Now it is: thermomechanical pretreatment; biogas; methane; biomass, lignocellulose materials
Introduction:
You didn't connect the paragraph on line 66 with the previous one. You start talking about biogas out of the blue.
Author’s reply: Thank you for your comment. The cross-reference between both paragraphs has been added: One of the ways of managing green energy is its use in agricultural biogas plants. Waste plant raw materials can be successfully used as a source of energy.
Line 74: remove extra space
Author’s reply: Thank you for your attention. The error has been corrected
Line 82: remove extra space
Author’s reply: Thank you for your attention. The error has been corrected
Line 84: Rapeseed should be capital letter
Author’s reply: Thank you for your comment. The error has been corrected
What are the objectives of your study? You must add a paragraph at the end of the introduction with this information. This is the basics of any paper.
Author’s reply: Thank you for your comment. A paragraph has been added to inform about the purpose of the work being carried out:
The aim of the research was to determine the influence of modification of the plasticizing system on the course of the extrusion-cooking process and on the physicochemical properties of selected pretreated straws as well as the effect of pretreatment on biogas efficiency during methane fermentation.
Materials and methods:
Line 129: How did you measure the moisture content? Which equipment was used?
Author’s reply: A moisture analyzer (Radwag 50.R.WH, Radom, Poland) was used to determine the initial moisture content of the straw.
Results and discussion:
Biogas and biomethane results should be presented in a Figure, not in a table. You don't need to show the results in 3 different units. Pick one and make a figure out of it. You will also be able to combine the results of extruded rapeseed straw, extruded rapeseed straw (after modification of the extruder screw), and extruded buckwheat straw (after modification of the extruder screw) in a single figure to allow comparison.
You must add standard deviations to all your results in the tables and in the figures your will create.
Author’s reply: Thank you for your comment. In Figures prepared with RSM method it is not possible to show standard deviation results required by the Reviewer.
As presented by: 37. Pilarski, K.; Pilarska, A.; Witaszek, K.; Dworecki, Z.; Å»elaziÅ„ski, T.; Ekielski, A.; Makowska, A.; Michniewicz, J. The impact of extrusion on the biogas and biomethane yield of plant substrates. Journal of Ecological Engineering 2016, 17, 264–272, https://doi:10.12911/22998993/64563, and by: 38. KozÅ‚owski, K., Mazurkiewicz, J., CheÅ‚kowski, D., Jeżowska, A., CieÅ›lik, M., Brzoski, M., SmurzyÅ„ska, A., Dongmin, Y., Wei, Q. The Effect of Mixing During Laboratory Fermentation of Maize Straw with Thermophilic Technology. Journal of Ecological Engineering, 2018,19, 93–98, https://doi.org/10.12911/22998993/91270, the determination of biogas and methane yields should be calculated on fresh matter, dry organic matter, on total solids and volatile solids, as necessary for compare of energetic potential between substrates. So, Authors prefer to show all the results as presented now in Tables 6-9 instead of figures required by the Reviewer. If figures would be prepared the huge amount of data to compare methane efficiency before and after modification depend on extrusion conditions and straw type made figure unreadable well. Results presented in Tables show all types of methane before and after plasticizing system modification and differences are quite visible.
The following sentence was added to the main text: The determination of biogas and methane yields, when calculated on fresh matter, dry organic matter or in total solids and volatile solids, is necessary for compare of energetic potential between substrates [37,38].
Reviewer 3 Report
There are some details that need to be clearer during the presentation of the paper.
The reader does not know what to expect at the beginning and it is important to present what you want to be presented in the paper from the beginning.
It is not explained from the beginning why straw extrusion is needed under certain conditions!
The reader (specialist) must be convinced of the efficiency of the extrusion before turning the straw into biogas!
There are some harder figures to read!

Author Response
There are some details that need to be clearer during the presentation of the paper.
The reader does not know what to expect at the beginning and it is important to present what you want to be presented in the paper from the beginning.
It is not explained from the beginning why straw extrusion is needed under certain conditions!
The research was carried out in order to increase the quality (break the lignocellulosic bonds) of the material processed in the biogas plant and thus increase the yield of biogas.
The reader (specialist) must be convinced of the efficiency of the extrusion before turning the straw into biogas!
There are some harder figures to read!
Thank you for your thorough analysis and pertinent comments that the Reviewer noticed.
- The abstract does not contain the synthesis of all the topics that are presented in the paper.
Author’s reply: Thank you for the suggestion. The following information has been added: It has been proved that the modification of the plasticizing system had a significant impact on the course of the process and the tested physicochemical properties. An important factor confirming the correctness of the modification is the increase in biogas efficiency. After modification the highest yield of cumulative biogas from fresh mass was 12.94% higher than in the sample processed before modification.
- Biogas is not discussed in the abstract
Author’s reply: Thank you for your opinion. information has been completed.
Was: …..efficiency of cumulative biogas and cumulative methane production expressed on dry mass, fresh mass, and fresh organic mass basis…
And added:
An important factor confirming the correctness of the modification is the increase in biogas efficiency. After modification the highest yield of cumulative biogas from fresh mass was 12.94% higher than in the sample processed before modification.
- The reader is not ready for what you are going to present!
Author’s reply: Thank you for your comment. A paragraph has been added to inform about the purpose of the work being carried out: The aim of the research was to determine the influence of modification of the plasticizing system on the course of the extrusion-cooking process and on the physicochemical properties of selected pretreated straws as well as the effect of pretreatment on biogas efficiency during methane fermentation.
- The figure does not distinguish the differences between the two extruder screws.
Author’s reply: Thank you for your opinion. information has been completed:
Added information: „(making additional cuts in the mixing zone – marked on Fig. 1 b.)”.
5.1 At what humidity were the experiments performed?
Author’s reply: The experiments performed were carried out using 20, 25, 30 and 35% moisture content.
5.2 Why are there differences between the extrudates of the two plants?
Author’s reply: Thank you for your comment. A paragraph has been added to indicate the content of certain chemicals: As the main raw materials rapeseed straw and buckwheat straw were used. The rape straw used in the tests was qualitatively assessed by: 4.76% ash content, 19.41% lignin content, 41.33% cellulose content, 30.57% hemicellulose content, 42.47% carbon content and 0.51% nitrogen content. In the case of buckwheat straw, there were: 5.73% ash content, 19.94% carbon content, 36.54% cellulose content, 28.26% hemicellulose content, 37.28% carbon content and 1.17% nitrogen content
5.3What was the initial degree of crushing?
Author’s reply: We added this information : „The raw materials used were shredded to a size of 8-10 mm…”
- What is the unit of measurement, actually? Why does '100%' appear?
Author’s reply: Thank you for your point. The mistake was corrected
- This is not stated in the abstract.
Author’s reply: The authors meant: ... ..efficiency of cumulative biogas and cumulative methane production expressed on dry mass, fresh mass, and fresh organic mass basis ...
- Pay attention to the resolution of the figures. Not all text is easy to read.
Author’s reply: Thanks for your comment. All figures have been checked and corrections made.
- What would be the percentage of lignin removed by extrusion in the two plants?
Author’s reply: Thank you for questions. After pretreatment the degree of cellulose crystallization and polymerization is reduced, and there is a larger specific surface area of the substrate which can be affected by microorganisms. Application of substrates pretreatment (e.g. extrusion), causing fragmentation of substrates, can give fermentation bacteria easier access to decomposable compounds by reduction of size of particles of the material to be processed. In consequence, the degree of cellulose crystallization and polymerization is reduced, and there is a larger specific surface area of the substrate which can be affected by microorganisms. Lignocellulose fibers contain lignins, which are hardly decomposable polymers. Cellulose biopolymers and hemicellulose can be easily decomposed by hydrolyzing bacteria. Lignocellulose fibers are the main building blocks of plant cell walls and lignin fraction is not removed by extrusion but it could be changed into soluble fractions of cellulose by its fragmentation after extrusion, as presented by Witaszek et al. [37] on Graphical abstract attached to the paper.
- Place the conclusions using bullets.
Author’s reply: The conclusions are presented with the help of paragraphs separating the individual results.
Reviewer 4 Report
This is an interesting topic. Please remove the weaknesses of this paper as following:
1. Many literatures have been reviewed in the paper, but less critical. In other words, you should clearly explain what contribution that has been made by former research and, in particular, what limitation /weakness that exists in each previous research. You should identify the gap of the knowledge from the review of the previous research to justify the significance of YOUR current research topic.
2. Abstract is very low. Clarify your novelty more in the abstract. Add your aims.
3. All figures and tables must be cited and described in the context. Check again.
4. Answer this question in the background, how you prove that this results are accurate; and do we use this conclusion for other applications too? how? How much do you assure of you results?
5. The conclusions should be presented and extended with more discussion regarding the results, and future development.
6. The authors should solicit additional proofreading from colleagues to further improve the readability of the document. the quality of figures should be improved for more readability.
7. Add the benefits and drawbacks of your work in conclusion. Add your future work too. Why do you do this?
Author Response
This is an interesting topic. Please remove the weaknesses of this paper as following
- Many literatures have been reviewed in the paper, but less critical. In other words, you should clearly explain what contribution that has been made by former research and, in particular, what limitation /weakness that exists in each previous research. You should identify the gap of the knowledge from the review of the previous research to justify the significance of YOUR current research topic.
Author’s reply: Has the bibliography been properly corrected and supplemented with a bibliography supplementing the state of knowledge
- Abstract is very low. Clarify your novelty more in the abstract. Add your aims.
Thank you for the suggestion. The following information has been added: It has been proved that the modification of the plasticizing system had a significant impact on the course of the process and the tested physicochemical properties. An important factor confirming the correctness of the modification is the increase in biogas efficiency. After modification the highest yield of cumulative biogas from fresh mass was 12.94% higher than in the sample processed before modification.
And in Introduction paragraph has been added to inform about the purpose of the work being carried out: The aim of the research was to determine the influence of modification of the plasticizing system on the course of the extrusion-cooking process and on the physicochemical properties of selected pretreated straws as well as the effect of pretreatment on biogas efficiency during methane fermentation.
- All figures and tables must be cited and described in the context. Check again.
Author’s reply: Thank you for your comment. According to the reviewer's suggestion, it was checked and then referred to the other tables and figures
- Answer this question in the background, how you prove that this results are accurate; and do we use this conclusion for other applications too? how? How much do you assure of you results?
Author’s reply: The correctness of the results was presented by means of the function of fitting the response surface model for the examined trait, standard deviations and by means of the PCA analysis.
- The conclusions should be presented and extended with more discussion regarding the results, and future development.
Author’s reply: The authors mentioned it in the conclusions section: In future research attempts, more attention should be paid to the proper selection of process variables for specific raw materials. It would be reasonable to carry out a comparative analysis using various extruder plastification unit configuration (L/D) or screw lengths, as well as measuring the influence of modification of the plasticizing system on the extrusion-cooking process stability and on methane efficiency.
- The authors should solicit additional proofreading from colleagues to further improve the readability of the document. the quality of figures should be improved for more readability.
Author’s reply: Thank you for your comment. Appropriate corrections were made to increase the legibility of the drawings, including the enlargement of the markings in Figure 14.
- Add the benefits and drawbacks of your work in conclusion. Add your future work too. Why do you do this?.
Author’s reply: According to the authors, the information contained in the conclusions section: "The following conclusions were drawn based on the conducted tests and the analysis of results. The results in individual tests differed depending on the raw material used. Not only may this be attributed to the structure of the tested raw material, but it may also suggest the need to select appropriate process parameters for a specific material. ... "and" In future research attempts, more attention should be paid to the proper selection of process variables for specific raw materials. It would be reasonable to carry out a comparative analysis using various extruder plastification unit configuration (L / D) or screw lengths, as well as measuring the influence of modification of the plasticizing system on the extrusion-cooking process stability and on methane efficiency. " refers to work defects and refers to future work.
Round 2
Reviewer 2 Report
All my comments have addressed.
Reviewer 4 Report
The authors have made considerable changes in the revised version of the manuscript. The manuscript is accepted in the present form.